# SELF-SUPERVISED IMAGE DENOISING WITH SWIN TRANSFORMER

**Pavel Chizhov, Mikhail Papkov**
Institute of Computer Science, University of Tartu
Tartu, Estonia
`{pavel.chizhov,mikhail.papkov}@ut.ee`

## ABSTRACT

Self-supervised image denoising aims to reconstruct signal from a noisy image with no additional information. Typically, this is accomplished by means of specific frameworks built upon fully-convolutional neural networks. In two such frameworks, Noise2Self and Noise2Same, we replaced conventional convolutional backbones with a state-of-the-art Swin Transformer-based model. In this paper, we summarize the results of experiments on a range of datasets and examine the advantages and limitations of transformers in self-supervised denoising.

## 1 INTRODUCTION AND RELATED WORK

Deep learning solutions for *self-supervised* denoising are usually based on a fully-convolutional U-Net architecture (Ronneberger et al., 2015). It was used in Noise2Void (Krull et al., 2019) and Noise2Self (Batson & Royer, 2019), which concurrently introduced masked pixels reconstruction as a proxy task for blind-spot denoising. The main limitation of masking was the inability to look at pixels' own noisy albeit useful signal. Noise2Same (Xie et al., 2020) addressed this issue with a dual forward pass of masked and unmasked images through the network.

SwinIR (Liang et al., 2021) is a state-of-the-art model for *supervised* image restoration, including denoising, based on Swin Transformer (Liu et al., 2021). Despite the superiority of this architecture, there was limited research exploring its capabilities in the self-supervised setting. DnT (Liu et al., 2022) utilized a variant of SwinIR for *unsupervised single-image* denoising, following R2R (Pang et al., 2021) to sample pairs of images with independent noise. In this work, we propose a simpler approach to validate SwinIR performance for self-supervised denoising by directly integrating it into Noise2Self and Noise2Same frameworks. Additionally, we conduct an ablation study to determine optimal hyperparameters on a range of datasets.

## 2 METHODS

Noise2Self uses a blind-spot loss — a mean squared error (MSE) between predicted and hidden values of masked pixels. Noise2Same makes an additional run through the model when the unchanged image is passed as input. In this case, the loss consists of two components: reconstruction MSE between the second output and the noisy input and invariance MSE between the two outputs computed over the pixel mask. The difference between frameworks is illustrated in Figure 2.

SwinIR model (Figure 3) consists of six groups with six Swin Transformer blocks in each group, followed by a $3 \times 3$ convolution. Two consequent blocks alternate between plain and shifted window multi-head self-attention. We used window size $16 \times 16$ and patch size $1 \times 1$ pixels, and did not use the global shortcut (adding input to output). We further discuss training details in Appendix B.

## 3 EXPERIMENTS AND DISCUSSION

The models were trained and evaluated on three noise datasets: BSD68 — grayscale images of natural objects (Martin et al., 2001), HanZi — grayscale images of handwritten Chinese characters (Batson & Royer, 2019), and ImageNet — RGB images of natural objects (Deng et al., 2009; Xie et al., 2020). We followed Xie et al. (2020) for training settings and data preparation, except for BSD68, where we used crops of size $128 \times 128$ and batch size 32 instead of $64 \times 64$ and 64.

Table 1: Denoising results. The best scores are highlighted in bold, and the second-best scores are underlined. We report the results of our implementation wherever possible and compare them to those reported in Noise2Same (Xie et al., 2020). † — clipping gradient norm to $1.0$.

| Method | Backbone | BSD68 | | HanZi | | ImageNet | |
| --- | --- | --- | --- | --- | --- | --- | --- |
| | | PSNR | SSIM | PSNR | SSIM | PSNR | SSIM |
| Noise2Void | U-Net (Xie et al., 2020) | 27.71 | — | 13.72 | — | 21.36 | — |
| Noise2Self | U-Net (Xie et al., 2020) | 26.98 | — | 13.94 | — | 20.38 | — |
| | U-Net | 26.88 | 0.734 | 14.16 | 0.512 | 21.33 | 0.574 |
| | SwinIR | 27.57 | 0.774 | 13.57 | 0.475 | 20.28† | 0.524† |
| Noise2Same | U-Net (Xie et al., 2020) | 27.95 | — | 14.38 | — | 22.26 | — |
| | U-Net | **28.11** | 0.781 | **14.85** | **0.542** | 22.85 | 0.625 |
| | SwinIR | 28.07 | **0.786** | 14.35 | 0.513 | **23.05** | **0.635** |

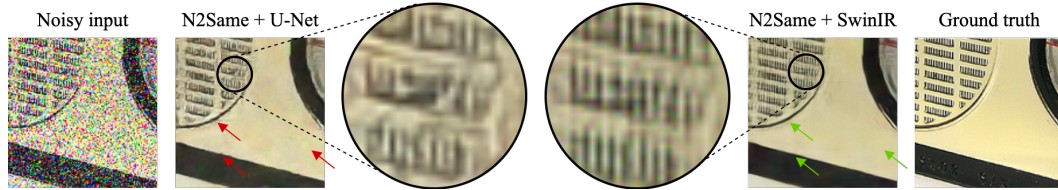

Figure 1: Image restoration results of U-Net and SwinIR in Noise2Same self-supervised framework.

The evaluation results are given in Table 1. SwinIR backbone improved the peak signal-to-noise ratio (PSNR) and structural similarity index (SSIM) in several experiments, such as Noise2Same on ImageNet (+0.20 PSNR) or Noise2Self on BSD68 (+0.69 PSNR). In Noise2Same on BSD68, SwinIR has a slightly lower PSNR but better SSIM. We hypothesize that SwinIR is more capable of detailed structure restoration than U-Net. In Figure 1, high-frequency details are better retained in the output of SwinIR. At the same time, in the picture produced by U-Net, the structure is blurry and uneven. As an exclusive case, however, HanZi dataset has a clear black background and less high-frequency signal than natural images — SwinIR did not improve its scores.

Being more computationally expensive than U-Net, SwinIR has its limitations regarding resources and processing time. We document the details in Appendix C, Tables 3 and 4.

**Ablations.** By default, SwinIR (Liang et al., 2021) embeds patches of size $1 \times 1$ px instead of $4 \times 4$ originally proposed in Swin Transformer (Liu et al., 2021). While this is an essential change for the image restoration task, it implicitly reduces window resolution from $28 \times 28$ to $7 \times 7$. Such window size is not computationally efficient, because input should be padded for divisibility (e.g., crop $64 \times 64$ will increase in size by 19.6%). Also, it may be too small for the image restoration task. We compared window sizes $8 \times 8$ and $16 \times 16$ and found that the larger one consistently improves the scores (see Table 6). We have also investigated the importance of larger crop sizes against larger batch sizes and found that crops of size $128 \times 128$ with batch size 32 are better than $64 \times 64$ and 64 on BSD68 for all backbones and frameworks except U-Net in Noise2Self (see Table 5).

## 4 CONCLUSION

We replaced a fully-convolutional architecture with a transformer-based model in self-supervised image denoising frameworks and assessed the results. Our experiments showed that SwinIR demonstrates competitive performance, being capable of restoring complex high-frequency details ignored by its convolutional counterparts. However, such improvement comes at a considerable computational cost. We did not experiment with 3D data, and it is yet unclear if it is feasible to denoise it with transformers. we leave this question open for further work.

URM STATEMENT

The authors acknowledge that at least one key author of this work meets the URM criteria of the ICLR 2023 Tiny Papers Track.

ACKNOWLEDGEMENTS

The authors thank Revvity, Inc. for the financing of this work and High Performance Computing Center at the Institute of Computer Science of the University of Tartu for providing computational resources.

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

# A  ARCHITECTURES

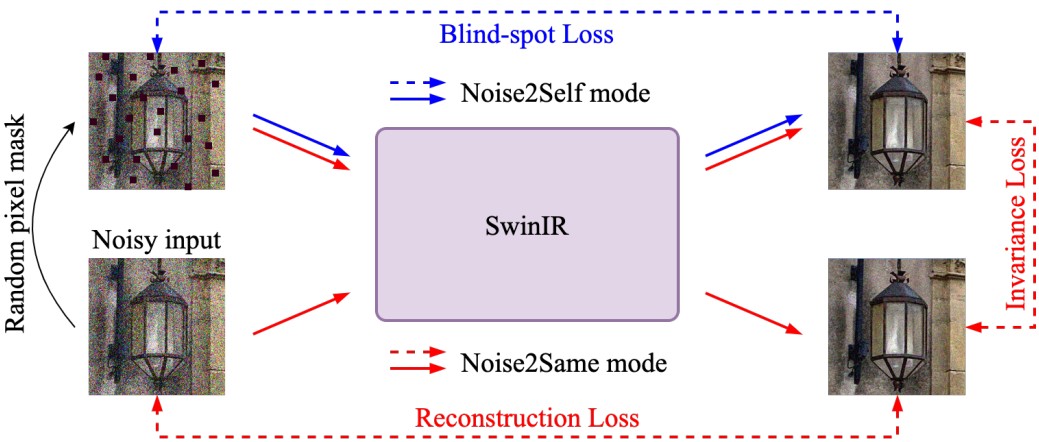

Figure 2: Noise2Self (Batson & Royer, 2019) and Noise2Same (Xie et al., 2020) frameworks.

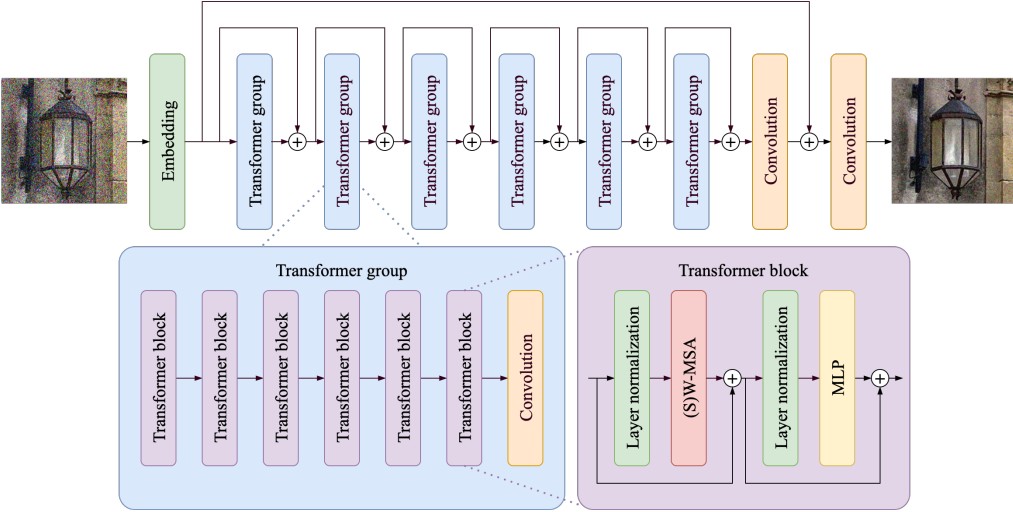

Figure 3: SwinIR (Liang et al., 2021) model architecture.

## B    TRAINING DETAILS

We used Adam optimizer (Kingma & Ba, 2014) in all experiments with initial learning rate of $4 \cdot 10^{-4}$ and scheduled learning rate decay by half every $5000$ iterations. The main training setup parameters are demonstrated in Table 2. We encountered a problem of exploding gradients in Noise2Self + SwinIR on HanZi with window size of $8 \times 8$ and on Imagenet. In these cases, we had to use gradient norm clipping of 1.0.

We implemented all models in Python 3.8.3 and PyTorch 1.12.1 (Paszke et al., 2017), and trained them on NVIDIA A100 80GB GPUs (driver version: 470.57.02, CUDA version: 11.4). We used `einops` (Rogozhnikov, 2022) library to implement transformer blocks in a verbose way.

Table 2: Training configuration.

|                    | BSD68 | HanZi | ImageNet |
|--------------------|-------|-------|----------|
| Batch size         | 32    | 64    | 64       |
| Training crop size | 128   | 64    | 64       |
| # iterations       | 80000 | 50000 | 50000    |

## C    COMPUTATIONAL COSTS

SwinIR is a more computationally expensive model than U-Net, so it was trained longer on multiple GPUs (see Tables 3 and 4 for details).

Table 3: Number of GPUs utilized in each experiment.

| Method     | Backbone | BSD68 | HanZi | ImageNet |
|------------|----------|-------|-------|----------|
| Noise2Self | U-Net    | 1     | 1     | 1        |
|            | SwinIR   | 4     | 2     | 2        |
| Noise2Same | U-Net    | 1     | 1     | 1        |
|            | SwinIR   | 8     | 4     | 4        |

Table 4: Training time (TT) and average inference time (AIT) in experiments. Inference was done in a single thread with batch size 1, as for BSD68 and ImageNet it is infeasible to compose batches due to varying image sizes.

| Method | Backbone | BSD68 TT (h) | BSD68 AIT (ms) | HanZi TT (h) | HanZi AIT (ms) | ImageNet TT (h) | ImageNet AIT (ms) |
|--------|----------|--------------|----------------|--------------|----------------|-----------------|-------------------|
| Noise2Self | U-Net  | 1  | 7    | 0.5 | 5  | 1  | 15    |
|            | SwinIR | 20 | 2632 | 12  | 40 | 14 | 11565 |
| Noise2Same | U-Net  | 2  | 8    | 1   | 6  | 1  | 14    |
|            | SwinIR | 26 | 3548 | 22  | 57 | 13 | 5985  |

# D ABLATION STUDY

Table 5: BSD68 training crop and batch size ablation study. The scores are compared within each method and backbone combination. The best scores are highlighted in bold.

| Method | Backbone | Crop size | Batch size | PSNR | SSIM |
|---|---|---|---|---|---|
| Noise2Self | U-Net | 64 | 64 | **26.94** | **0.743** |
| | | 128 | 32 | 26.88 | 0.734 |
| | SwinIR | 64 | 64 | 27.49 | 0.769 |
| | | 128 | 32 | **27.57** | **0.774** |
| Noise2Same | U-Net | 64 | 64 | 27.90 | 0.752 |
| | | 128 | 32 | **28.11** | **0.781** |
| | SwinIR | 64 | 64 | 27.78 | 0.736 |
| | | 128 | 32 | **28.07** | **0.786** |

Table 6: Window size ablation study. The scores are compared across window sizes of SwinIR in each method separately. The best scores are highlighted in bold. $\dagger$ — clipping gradient norm to $1.0$.

| Method | Window size | BSD68 | | HanZi | | ImageNet | |
|---|---|---|---|---|---|---|---|
| | | PSNR | SSIM | PSNR | SSIM | PSNR | SSIM |
| Noise2Self | 8 | 27.54 | 0.772 | $12.18^{\dagger}$ | $0.388^{\dagger}$ | $19.41^{\dagger}$ | $0.470^{\dagger}$ |
| | 16 | **27.57** | **0.774** | **13.57** | **0.475** | $20.21^{\dagger}$ | $0.493^{\dagger}$ |
| Noise2Same | 8 | 28.00 | **0.786** | 14.24 | 0.510 | 22.95 | 0.626 |
| | 16 | **28.07** | **0.786** | **14.35** | **0.513** | **23.05** | **0.635** |

# E    ADDITIONAL EXAMPLES

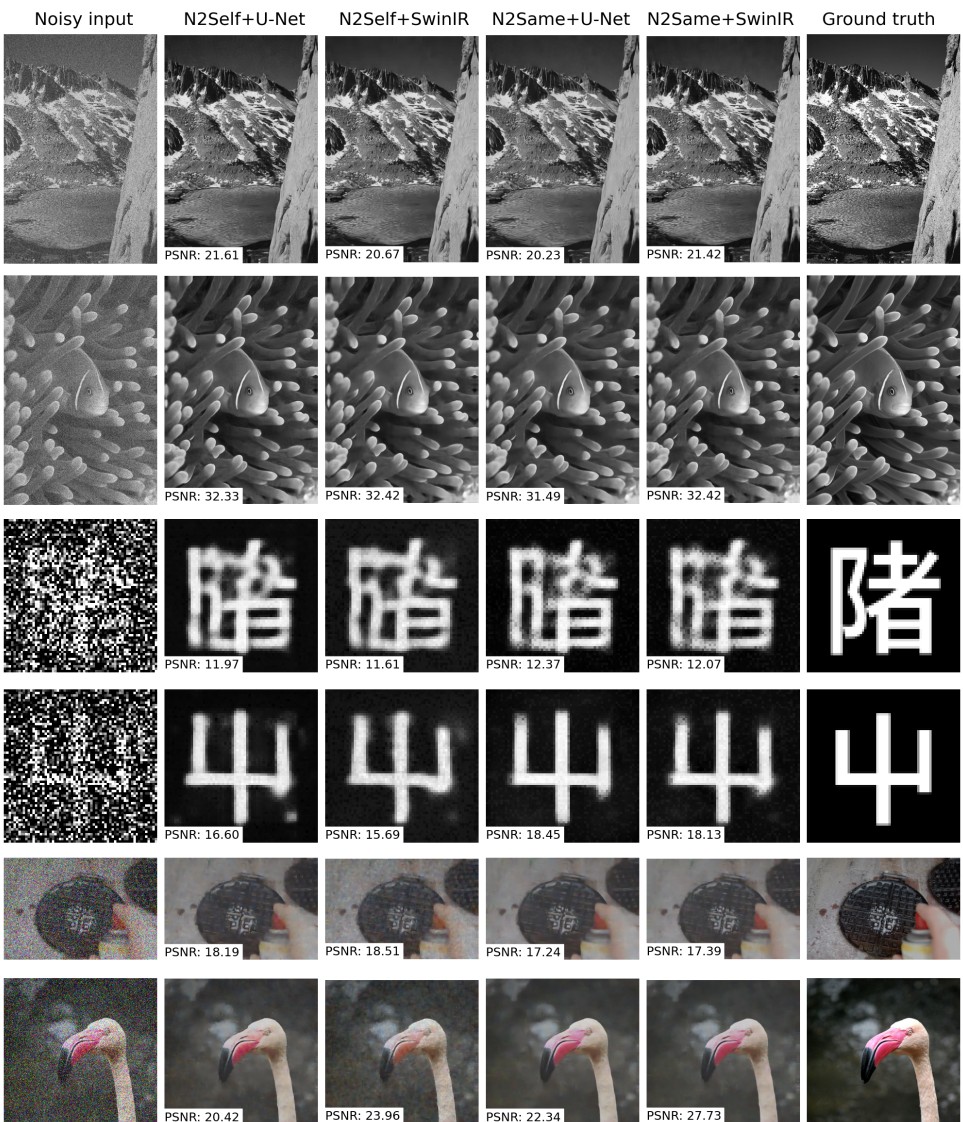

Figure 4: Additional examples of images denoised in BSD68 (first pair of rows), HanZi (second pair of rows), and ImageNet (last pair of rows) experiments. Each pair of rows contains a hard and a simple example from the dataset with the corresponding PSNR scores.

