# OpenReview forum: "Self-Supervised Image Denoising with Swin Transformer"
_ICLR.cc/2023/TinyPapers — Submitted to Tiny Papers @ ICLR 2023_

### Official Review · Reviewer_cN8B · 2023-03-30

**Confidence:** 4

**Summary Of Contributions:**

This paper study the usefulness of Swin transformer in self-supervised image denoising by replacing the fully convolutional blocks with the Swin transformer block. The authors tested the proposed approach with two image denoising methods and on three different datasets.

**Rating:**

Clear, Correct, and Reproducible (CCR): a submission which meets the reviewing criteria

**Strengths And Weaknesses:**

- Clarity: The paper is written clearly and relevant literature is discussed.

- Correctness: As claimed, the paper discussed the advantages of using Swin transformers over convolution block and discussed its limitations in terms of throughput and resources.
    - I am concerned about the usefulness of the proposed approach as table 1 shows mixed results with SwinIR as the backbone.
    - Given that SwinIR as backbone requires lots of resources as compared to convolution network how one would justify its use for the methods where SwinIR performance is lower than U-Net?
    - Further, the performance gain of SwinIR over U-net is very small, making me doubt the usefulness of SwinIR.

- Reproducibility: No code and data is provided

- Follows basic requirements: yes

**Suggested Changes:**

- Experiments:
    - Performance statistical significance test to better understand if the slight performance gain with SwinIR not because of a random chance.
    - I would also reason about why the proposed change does not work for all three datasets.

---

> ### Author Response · Authors · 2023-06-01
> **Comment for the review**
>
> Thank you very much for the feedback. We are pleased that our paper appears clearly written and glad that the discussed literature is considered relevant.
>
> We agree that SwinIR as a backbone shows mixed results while demanding computational resources. This is the main limitation of the approach that we determined in the work. However, we have also discovered the difference in texture restoration (Figure 1, and the general difference in SSIM scores), where SwinIR showed greater capability. Considering this, the justification for using a transformer-based backbone should be made in terms of the cruciality of fine textures to one's domain. For example, in biomedical image analysis, the smoothing effect of a CNN-based backbone could lead to considerable changes in the image signal. This could likely hinder the practitioner's analysis of the data.
>
> Regarding the reason why the proposed change didn't improve the scores on all three datasets, we formulated our assumption about the HanZi data in the paper:
> > HanZi dataset has a clear black background and less high-frequency signal than natural images –  SwinIR did not improve its scores.
>
> We can add that the reason for the performance on the BSD68 data may lie in the dataset size and diversity – it it orders of magnitude smaller than ImageNet.
>
> We are grateful for the insightful feedback that allows us to improve our work in the future.

---

### Official Review · Reviewer_mQtQ · 2023-04-01

**Confidence:** 5

**Summary Of Contributions:**

this paper proposes improving the self-supervised denoising in 2 frameworks (noise2same and noise2self) by replacing the convolutional backbones with a swin transformer model (swinIR). This work is validated on 3 publicly available datasets - bsd68, HanZi and ImageNet

**Rating:**

Great Start (GS): a submission which meets some of the reviewing criteria but has room for improvement

**Strengths And Weaknesses:**

a) Clarity: comments below
Strengths:

- Detailed experiments and ablation studies have been performed on a set of publicly available datasets with varying sizes.
- It provides a helpful reference to assess the computational costs involved while using a transformer-model versus conventional convolutional networks (u-net)
- the paper is well-structured though there is some room for improvement in the methods and conclusion sections.

b) Correctness: some comments/discussion points that I am raising here

1. noise2same: as pointed out by the authors, unet outperforms swinir on the hanzi dataset. for hanzi dataset, did the authors try adding more noise or blurring to see if reduction in high-frequency content leads to any changes in the results ?
Related: I am asking this because the authors conclude that 'SwinIR is capable of restoring complex high-frequency details' - what kind of complexity are you referring to here ?

2. noise2self: for both hanzi and imagenet, noise2same (xie, 2020) and unet outperforms swinIR (both psnr and ssim). Would the authors hypothesize why this is the case (at least for imagenet) when the results are the opposite for the bsd68 dataset ?

3. As a discussion, do the authors think that there is 'sweet spot' in terms of the psnr-ssim tradeoff that can be considered as a guideline (example: beyond a certain increase in psnr, the ssim begins to deteriorate). If yes, then it would be great to have this in the paper.

4. do the authors feel that given the significant computational overhead involved vs the minor changes, it is feasible to use swin transformers for these particular cases ? (also please look at the references given in the 'suggested changes' section)

c) Reproducibility: Code not provided. The datasets used are publicly available.

d) Follows basic requirements: Yes


**Suggested Changes:**

1. Please address the points raised in the previous section (correctness)

2. figure 1 has been supplemented with additional examples in figure 4. It would have been nicer to have a few regions zoomed in (as done in fig.1) to better visualize the extent of smoothing and the retaining of structural similarity

3. to modify the conclusions section: for feasibility of using transformers for 3d data, please take a look at the following papers (which I guess you may already have):
a. https://openaccess.thecvf.com/content/CVPR2022/papers/Tang_Self-Supervised_Pre-Training_of_Swin_Transformers_for_3D_Medical_Image_Analysis_CVPR_2022_paper.pdf

b. https://www.cell.com/patterns/pdf/S2666-3899(22)00083-6.pdf

If addressed suitably, I would then increase the score assigned to CCR.

---

> ### Author Response · Authors · 2023-06-01
> **Comment for the review**
>
> We are grateful for the verbose and insightful feedback. We are glad that our paper’s experimental setup and ablations appear well-detailed. Also, we are pleased that our overview of computational costs is considered helpful. We are happy to address the discussion points in this reply.
>
> > did the authors try adding more noise or blurring to see if reduction in high-frequency content leads to any changes in the results?
>
> We tried HanZi on several noise levels, though our approach was equally inferior to U-Net in all cases, so we decided not to report the redundant results. Experimenting with blur seems an interesting direction, except that the noise levels we are working on are already very heavy and Noise2Same is not reported to work for de-blurring as-is.
>
> Regarding the performance in the Noise2Self setting, the reason for SwinIR drastically losing in performance compared to other methods is likely in the severe gradient clipping in these experiments – otherwise, the training didn't converge at all. We hypothesize, that the stochastic blind-spot loss of Noise2Self computed on a small partition of pixels is harmful to complex transformer models.
>
> > do the authors think that there is 'sweet spot' in terms of the psnr-ssim tradeoff that can be considered as a guideline?
>
> We believe that each data case should be considered separately due to the different nature of each dataset. The best way would be to examine the model outputs and see, which values are acceptable for a certain use case.
>
> > do the authors feel that given the significant computational overhead involved vs the minor changes, it is feasible to use swin transformers for these particular cases?
>
> In terms of the practical applicability of the presented approach, one should judge if the computational overhead is negligible compared to the performance benefits a transformer model has the potential to bring. Those being texture restoration and less amount of smoothing critical to some domains, e.g., biomedical imaging.
>
> We are especially thankful for the suggested changes. Though we have not yet started to investigate the 3D case, the provided papers are highly essential to our future work.

---

### Meta-Review · Area_Chair_2i9K · 2023-04-08

**Recommendation:** Invite to archive
**Confidence:** 4

**Metareview:**

Based on the two reviews provided, the paper proposes replacing convolutional backbones with Swin transformers for self-supervised image denoising in two frameworks (noise2same and noise2self). The proposed approach is tested on three publicly available datasets: bsd68, HanZi, and ImageNet.

Review 1 acknowledges the detailed experiments and ablation studies performed on the datasets, and the paper's clear structure. However, the review raises concerns about the correctness of some of the conclusions, such as why SwinIR performs worse than U-Net on some datasets, and whether there is a tradeoff between PSNR and SSIM. Review 1 also questions the feasibility of using Swin transformers given their significant computational overhead compared to minor improvements in results.

Review 2 agrees with the clarity and relevance of the paper, but raises concerns about the usefulness of the proposed approach. Table 1 shows mixed results, and the performance gain of SwinIR over U-Net is small. Additionally, SwinIR requires significantly more resources than convolutional networks, which raises concerns about justifying its use when its performance is lower than U-Net.

In summary, the paper's strengths include its clear structure and detailed experiments and ablation studies. However, concerns have been raised about the correctness and usefulness of the proposed approach, particularly regarding the tradeoff between performance and computational resources. The main message of the paper is that Swin transformers can be used as backbones for self-supervised image denoising, but their benefits may be limited and come at a cost of increased computational resources.


**Summary:**

This paper study the usefulness of Swin transformer in self-supervised image denoising by replacing the fully convolutional blocks with the Swin transformer block.

**Reason For Not Giving A Higher Recommendation:**

 It can be further improved when incorporating the suggested changes by the reviewers.


**Reason For Not Giving A Lower Recommendation:**

Based on the review comments from both reviewers, I think this submission does provide something new and meaningful to the field.

---

### Decision · Program_Chairs · 2023-04-09

Invite to archive

---

> ### Author Response · Authors · 2023-05-31
> **Opt-in for archival**
>
> We would like to inform that we have uploaded the updated version of the paper and we wish to opt for the archival

---

> ### Comment · Area_Chair_2i9K · 2023-06-06
> **Meet threshold for archival**
>
> This work meets the threshold for archival, contents the URM statement and is deanonymized.